# Case Report-Driven Medical Education in Rural Family Medicine Education: A Thematic Analysis

**DOI:** 10.3390/healthcare11162270

**Published:** 2023-08-11

**Authors:** Ryuichi Ohta, Chiaki Sano

**Affiliations:** 1Community Care, Unnan City Hospital, 699-1221 96-1 Iida, Daito-cho, Unnan 699-1221, Japan; 2Department of Community Medicine Management, Faculty of Medicine, Shimane University, 89-1 Enya cho, Izumo 693-8501, Japan; sanochi@med.shimane-u.ac.jp

**Keywords:** case report, case-based education, community hospital, family medicine, rural, VUCA, collaboration, researcher

## Abstract

Case-based education (CBE) is a teaching method in which learners work on real-life cases to learn and apply concepts and skills they have been taught. Case report-driven medical education using the CBE framework can effectively facilitate student and resident learning, and entice them to become involved in actual clinical practice. Specific case report-driven medical education methods and learning outcomes are not clarified. This study aimed to clarify the specific learning processes and outcomes of case report-driven medical education in rural community-based medical education. Using a qualitative design based on a thematic analysis approach, data were collected through semi-structured interviews. The study participants were medical students and residents in training at a rural Japanese community hospital. Fifty-one case reports were completed and published in Cureus from April 2021 to March 2023. Participants learned about various difficulties related to volatility, uncertainty, complexity, and ambiguity (VUCA) in the medical care of various older patients, which increased their interest in family medicine. They appreciated the importance of case reports in academic careers and how their responsibilities as researchers increase with collaboration. Case report-driven medical education in community hospitals can drive medical students’ and junior residents’ learning regarding family medicine in the VUCA world.

## 1. Introduction

Case-based education (CBE) is a teaching method in which students work on real-world cases to learn and apply concepts and skills they have been taught [1]. CBE can be performed in undergraduate and postgraduate medical education [2,3], and allows learners to gain hands-on experience and develop critical thinking and problem-solving skills [4,5]. CBE is often used in clinical activities such as research, case studies, and simulations [6,7,8]. This method aims to prepare learners for the challenges they will face in real clinical situations and to enhance their understanding of clinical medicine [9,10].

CBE can be challenging for both students and educators owing to several limitations. One limitation is time management. Cases can take time and learners may struggle to balance their case work with other responsibilities, such as studying for examinations [11,12]. A lack of structure can detrimentally affect learning [13]. Cases can be open-ended, and learners may have difficulty continuing CBE, causing confusion and frustration [14]. Access to specialized equipment or facilities for cases may be limited, impeding learning [15]. CBE cases require collaboration; collaboration skills can influence the effectiveness of CBE [16]. For example, a lack of effective collaboration may limit feedback from fellow members. Accessing real patients can also be a limitation due to ethical and practical considerations [17]. For effective CBE, cases should incorporate time limitations, clear products, an efficient structure, efficient usage of limited resources, effective collaboration among members, continual feedback, and involvement in actual patient care [18].

An essential part of CBE involves writing case reports for assigned patients in clinical rounds. Writing case reports is one of the fundamental research skills physicians require to demonstrate professionalism [19]. To write case reports, physicians must assess patients’ conditions for diagnosis and identify their symptoms to assess the effectiveness of treatments [20,21]. Medical students and residents can learn the real process of diagnosis and treatment by writing case reports with the support of their supervisors and teachers [22]. In writing, they can effectively discuss and collaborate with patients, colleagues, and supervisors [23].

Case report-driven medical education within the CBE framework can effectively drive student and resident learning, and increase their involvement in actual clinical practice. In medical education, the involvement of medical trainees in clinical situations can be integral for better learning. Cognitive apprenticeship (CA) and legitimate peripheral participation (LPP) effectively involve medical trainees [24,25]. These frameworks allow trainees to gradually increase the extent of their participation in clinical medicine [24,25]. Writing case reports with supervisors or medical teachers and examining patients together may motivate trainees to participate in clinical medicine [22]. Sequential methods of writing case reports, accompanied by gradual exposure to medical examination of patients, are essential for case report-driven medical education concerning CA and LPP [24,25,26]. Writing case reports may also allow supervisors and medical teachers to reflect on patient care [27]. Case reports do not require many resources and can be completed through discussions between trainees, supervisors, and teachers [27].

Case report-driven medical education is particularly useful in resource-limited medical education, such as rural community-based medical education (CBME), which lacks medical teachers and teaching resources [28,29,30]. In addition, learning through writing case reports of complicated older patients within the educational framework of CA and LPP in rural contexts could enhance the understanding of various older patient presentations. However, the learning processes and outcomes of case report-driven medical education have not been clarified in current medical education.

CBE using older patients’ medical issues improves the understanding of polypharmacy and multimorbidity and approaches dealing with these issues through interprofessional collaboration. However, learning practical clinical courses and treatments of older patients’ care with the realization of reality can be challenging in present CBE without concomitant actual patient experience [28,29,30]. Case report-driven medical education can include older patients’ medical issues and authentic experiences, driving their learning and preparation for actual work. In this regard, clarifying the learning processes and outcomes of specific case report-driven medical education in resource-limited settings can motivate medical educators in rural contexts to adopt CBE and case reports, thereby benefiting rural CBME. To this end, this study posed the following research question: “What and how do medical students and residents learn about older patients’ care and physicians’ academic careers through case report-driven medical education in rural community hospitals?” Thus, it aimed to clarify the specific learning processes and outcomes of case report-driven medical education in rural CBME.

## 2. Materials and Methods

### 2.1. Setting

The study was conducted at the Unnan City Hospital in the southeast Shimane prefecture in rural Japan. The hospital had 281 care beds, of which 160 were for acute care, 43 for comprehensive care, 30 for rehabilitation, and 48 for chronic care. The hospital provided CBME to medical students and residents of medical universities and tertiary hospitals. Under this curriculum, medical students and residents experienced multiple clinical situations in treating their patients, such as hospital, outpatient, home, and community care [31]. Prior to joining the Unnan City Hospital, medical students and junior residents received rural family medicine education at medical universities and tertiary hospitals. They trained in family medicine at rural hospitals for a month with medical teachers and family medicine residents. It was mandatory for them to train at a rural hospital as part of their university or hospital curriculum. The range of practices allowed for medical students and residents, respectively, differed: medical students needed the observation of medical teachers in each encounter with patients, but residents could examine patients individually; then they had to discuss managing them with medical teachers before prescribing and ordering tests and specific procedures. The Unnan City Hospital accommodated 40–50 medical students and junior residents annually for training at the Department of General Medicine, with bedside teaching and constant reflection [32,33].

### 2.2. Participants

CBME education at the Unnan City Hospital accepted medical students and residents from anywhere in Japan. Between April 2021 and March 2023, 53 medical students and 16 junior residents participated voluntarily in the CBME curriculum, which included family medicine at the Unnan City Hospital. The training aimed to produce competencies in general medicine areas required by Japan, such as person-centered care, comprehensive and integrative approaches, interprofessional work, community orientation, professionalism, and systematic practice [34,35]. Ultimately, 45 medical students, 7 junior residents, and 8 family medicine residents agreed to write case reports during their training and participation in this study. We used purposive sampling to address the research purposes for observation, field notes, and interviews. Data were collected using field notes and semi-structured interviews to investigate the concrete experience and learning of the participants.

### 2.3. Case Report-Driven Medical Education

Case report-driven medical education began in the first week of rural CBME training, during which a family medicine teacher explained case report-driven medical education to the learners.

Medical students examined cases with family medicine residents and educators, taking clinical histories and noting physical examinations. Based on these clinical notes, they discussed patients’ clinical conditions. In some cases, medical educators provided several articles and learning materials to medical students and junior residents to facilitate their understanding of patients’ conditions. Based on the discussions and relevant reading material, medical students and residents progressed in writing their case reports. Cases of patients actively infected with COVID-19 were not allowed to participate in this study because of the risk of infections and the limitation of interaction with patients.

The case reports comprised four sections: introduction, case presentation, discussion, and conclusions. The writing was facilitated by discussions between medical students, junior residents, family medicine residents, and teachers. Case selection was performed through discussions with the participants. First, medical students and junior residents summarized patients’ diseases and specific features. Residents and teachers of family medicine reviewed and revised those descriptions. Second, the medical students and junior residents described the patients’ clinical histories, physical examinations, specific tests, and treatments. When the medical students and junior residents struggled with their descriptions, they discussed them with family medicine residents and teachers. They re-examined the patients with family medicine residents or medical teachers to supplement their documentation.

Third, the medical students and residents described the discussion and conclusions based on what they had learned. Through discussions with family medicine residents and teachers, they selected topics and started to draft their discussion sections. Family medicine residents and teachers reviewed the medical students’ descriptions and revised the content regarding paragraphs and academic writing. All documentation for the case reports was provided by medical students, junior residents, family medicine residents, and medical teachers. Finally, family medicine residents and medical teachers reviewed and revised the content of the case reports and submitted each manuscript to international medical journals, mainly Cureus.

### 2.4. Data Collection

This was a qualitative study using a thematic analysis approach. The first researcher conducted semi-structured interviews with medical students, junior residents, and family medicine residents who went through the process of writing case reports. The specialties of the researchers were family medicine, medical education, public health, and rheumatology. The first researcher acted as a participatory observer in this research and took field notes on the observations of the participants as they learned from writing the supporting case reports. Based on the field notes, the researcher reflected on the process and participants’ learning. The first researcher interviewed all participants in the conference room at the hospital during the observation period. The interview guide included four questions: What did you think of the experience of writing case reports? What did you learn through writing case reports with respect to patient management? What did you learn through writing case reports regarding physicians’ careers? Do you have any ideas on learning by writing case reports? Each interview lasted approximately 30 min. The semi-structured interviews and field notes regarding case report-driven medical education were used in the analysis.

### 2.5. Analysis

The thematic analysis was conducted to clarify the participants’ learning through case report-driven medical education [36]. All field notes and interviews were recorded and transcribed verbatim. After reviewing the field notes and conducting the first five semi-structured interviews, the first researcher, R.O., coded the data and developed codebooks based on the repeated reading of field notes as initial coding for reliability. This study used process and concept coding methods [37]. The coded contents during the initial coding were shared with the second researcher, C.S., and the two researchers discussed the contents for triangulation. Then, R.O. continued initial coding individually, creating tentative concepts. The interview contents were analyzed iteratively during the research period after finishing each participant’s case report for the purposes of theoretical saturation. After the initial coding and creating tentative concepts, R.O. induced, merged, deleted, and refined the tentative concepts and created themes by going back and forth between the research materials and initial coding for the second coding. During the code refinement process, second coding was used to elaborate on tentative concepts and themes. R.O. and C.S. discussed the concepts and themes created through axial coding for further triangulation. Through the discussion, the concept of VUCA (volatility, uncertainty, complexity, ambiguity) appeared, which was incorporated into the themes and concepts regarding the previous article on the present medical education [38]. Finally, concepts and themes were discussed by both researchers, who ultimately agreed on the final themes.

### 2.6. Ethical Considerations

Participants’ anonymity and confidentiality were ensured throughout the study. All participants completed informed consent forms in writing before participating in this research. All procedures in this study were performed in compliance with the Declaration of Helsinki, as amended. The Unnan City Hospital Clinical Ethics Committee approved the study protocol (no. 20210032).

## 3. Results

### 3.1. Demographic Data

A total of 45 medical students (six 4th-grade, 16 5th-grade, and 23 6th-grade participants), 7 junior residents, and 8 family medicine residents participated in this research. The female percentage was 43.3% (26/60). Finally, 51 case reports were completed from April 2021 to March 2023 and published in the international medical journal *Cureus* (Table 1). The described patients’ average age was 74.5 years (standard deviation = 18.1), and 47.1% were male.

### 3.2. Results of the Thematic Analysis Approach

The thematic analysis approach, using semi-structured interviews, clarified three theories regarding learning through case report-driven medical education in community hospitals. Three themes and 10 concepts appeared in the analysis (Table 2). The three themes regarding participants’ learning were perception of medicine, academic career, and responsibility in collaboration.

Regarding the perception of medicine, the participants learned about various difficulties in rural family medicine related to VUCA in the medical care of various elderly patients by comparing them with tertiary and university hospitals. Regarding academic careers, participants increased their interest in family medicine and realized the importance of case reports in furthering their academic careers. They also learned about the realistic implementation of evidence in their cases. Regarding responsibility in collaboration, their responsibility as researchers increased. Through collaboration, they realized the importance of scheduling to complete case reports with prompt communication with the co-author.

### 3.3. Perception toward Medicine

#### 3.3.1. Volatility in Geriatric Care

While managing and writing about the cases, the participants observed an acute exacerbation of the conditions of elderly patients they did not encounter in their previous learning situations. The participants had experienced younger patients’ medical care before attending the community hospital. One of the junior residents stated, “I have never experienced such older patients with various diseases. They were vulnerable to subtle changes in their circumstances, causing infections and delirium.” (Junior resident 3). The participants realized that elderly patients differed from patients of younger generations and needed special care. One of the residents stated, “Elderly patients can suddenly worsen during their medical care. They respond to standard care, but their renal and liver functions can change with subtle changes in homeostasis. The same treatment may not be easily applied to different patients. Family physicians frequently had to observe the symptoms of elderly patients, as opposed to less frequently with younger patients.” (Junior resident 5). The participants observed a difference between the elderly and younger generations regarding their reaction to treatments. By writing case reports, they deepened their understanding of the specialty of geriatric care.

#### 3.3.2. Uncertainty of the Clinical Course

The participants felt that the clinical course descriptions were complicated because of the uncertainty of the patients’ clinical courses. A huge gap existed between textbooks and elderly patients’ clinical courses at hospitals. They considered that elderly patients’ clinical courses were complicated and did not foresee them being effectively managed using only their knowledge of diseases. One participant recorded, “I was shocked to observe various changes in the conditions of elderly patients. During the description of the case reports, I learned that elderly patients’ conditions can easily change, and clinical courses are uncertain.” (Student 13). Other participants said the same thing and struggled to describe the clinical course in their case reports. Through their case reports, the participants understood that patient care at community hospitals was complicated and difficult to manage in certain respects.

#### 3.3.3. Complexity of Medicine

The participants were surprised by the number of clinical problems that their patients had during admission. Compared with the cases they had handled at universities or tertiary hospitals, patients in community hospitals were complicated, and participants initially struggled to describe their cases. One of the medical students stated, “My case is very complicated, and each medical problem of the case is interwoven with high complexity. I could not write my case report’s background and lost the focus of my case.” (Student 2).

The participants frequently discussed their cases with family medicine residents and teachers to find the focus of their case reports. Through the discussions, they could understand that the medical conditions between tertiary hospitals and community hospitals were different, and that the complexity of medicine should be considered in community hospital care. One participant said, “The discussion with family medicine residents and teachers facilitated my understanding of the complexity of elderly patients in the hospital and the reality of those complexities.” (Student 11).

#### 3.3.4. Ambiguity of Patients’ Decisions

The participants learned about the difficulty of making treatment decisions for elderly patients due to multiple factors affecting their decision-making. While writing the case reports, they noticed that decision-making in treating elderly patients at the community hospital differed from that of university and tertiary hospitals. At community hospitals, many dependent patients required support from their families in various ways. One of the participants remarked, “Elderly dependent patients could not choose their treatment by themselves. A number of key individuals influence patients’ decision-making.” (Junior resident 8).

In the discussion part of the case reports, the participants had to write about their cases based on evidence. However, they realized that elderly patients’ psychosocial factors made it challenging to apply evidence to the case. One of the participants shared that “Writing the discussion section was challenging. I initially felt that my case was not evidence-based and was treated appropriately.” (Student 4).

Through discussions with family medicine residents and teachers, participants gradually realized the ambiguity of patients’ decision-making and the reality of medicine in communities. One of the participants explained, “I could not understand why certain treatments were performed on my patient, based on the evidence. However, I realized through a discussion with residents and teachers that the evidence has huge limitations for application, especially for elderly patients. In reality, ambiguous decision-making should be respected in elderly dependent patients’ decision-making.” (Junior resident 1).

### 3.4. Academic Career

#### 3.4.1. Interest in Family Medicine

By writing case reports, the participants became interested in family medicine. Discussions with family medicine residents and teachers facilitated their understanding of the complicated conditions of their patients. One of the participants recorded the following experience, “Initially, the patient’s condition was so difficult for me. The patient’s condition is affected by multiple factors. Thanks to the support of residents and teachers, I could understand their relationships.” (Student 11). Another participant shared, “Continual discussion with family medicine teachers while writing my case report drove my interest in family medicine. I consider that understanding and solving the complicated problems of patients can be interesting points of family medicine.” (Student 5).

The challenges caused by the uncertainty and volatility of geriatric patients stimulated participants’ interest in family medicine. Participants could simultaneously use diverse medical knowledge and multiple medical skills, and experience a sense of accomplishment in clinical practice. One of the participants mentioned that “In family medicine, I needed to use various medical knowledge from different categories of medicine in one case. As it was difficult, I felt that family medicine was challenging for me.” (Junior resident 2).

#### 3.4.2. Importance of a Case Report in an Academic Career

Writing case reports through discussions revealed the importance of taking up academic careers in clinical situations. They believed that writing case reports made learning effective in medical science and enhanced their academic careers. One of the participants stated, “Writing case reports could effectively advance medical science. The clinical course in my case was different from that reported in the literature. My case report can inform other clinicians who have experienced similar cases. This is a kind of process of compiling evidence.” (Student 20). Through discussions with family medicine residents and teachers, participants realized that the residents and teachers learned a lot from writing case reports and improved their medical performance. One of the participants pointed out that “Writing case reports could stimulate physicians to learn more and describe their experience as evidence in medical science. Through discussions with the residents and teachers, I am motivated to write case reports in the future.” (Junior resident 10)

#### 3.4.3. Realistic Implementation of Evidence

The experience of writing case reports supports the learning of evidence implementation in complicated cases. While writing up discussions, participants realized that various scientific papers did not fit their cases. One of the participants disclosed, “In searching scientific papers, as I found various papers, there were differences between my case and the previous papers. Through writing the discussion up, I learned the limitations of the application of evidence in real cases.” (Junior resident 12). In addition, through discussions with the residents and teachers, participants learned about the limitations of the application of evidence because of patients’ preferences and social factors. One of the participants expressed the view that “Patients’ treatments are affected by their preferences and social factors, including their families. We as medical professionals have to respect such factors and handle them smoothly.” (Student 19).

### 3.5. Responsibility in Collaboration

#### 3.5.1. Responsibility as a Researcher

The participants felt the responsibility to complete the case reports. Through discussions with the residents and teachers, they needed to and could complete their roles in writing. One of the participants stated, “Finishing case reports requires a strict schedule. Each case report clarifies the news for other clinicians and researchers. I felt some responsibility to finish the case report.” (Student 16). The initial progress was slow because of the participants’ vague understanding of their role in writing case reports. Another participant said, “A clear role for each researcher may be essential to finishing the case reports. Initially, as a student, I could not realize the importance of writing case reports. However, after discussions with residents and teachers, I felt obligated to move forward with the case report.” (Student 8). Assigning clear roles to medical students is vital for the progression of writing case reports.

#### 3.5.2. Importance of Scheduling

The participants felt the importance of scheduling in writing case reports. One of the participants opined, “Having a role in a case was important for completing the case report because it could realize me a responsibility. For effective completion, I realized that scheduling was essential.” (Student 2). Another participant said, “When I got busy with other things, I tended to be reluctant to embark on writing the case report. However, the chosen schedule was effective, driving me to restart writing the case report.” (Junior resident 11). The participants realized that, for effective work in writing case reports, research, scheduling and adhering to the schedule were essential to remain organized.

#### 3.5.3. Prompt Communication with the Co-Author

While writing the case reports, participants experienced difficulties and sometimes stopped writing. At such times, some participants would promptly overcome difficulties by consulting residents and teachers. One of the participants reported, “I experienced some difficulties in writing the case report, especially about the contents of the introduction and discussion and academic writing. Although I tried to deal with them, I promptly consulted my teacher. The consultation facilitated my writing effectively.” (Junior resident 4). Prompt communication and discussion with residents and teachers could effectively solve problems. Through prompt communication, participants realized the importance of different perspectives for the effective progression of research. One participant advised, “I could progress with the case report by myself. However, my challenges should be conveyed because there is a time limit, and prompt advice from the residents and teachers can improve my writing regarding quality and speed of progression.” (Student 21).

## 4. Discussion

This study clarified the effects of case report-driven medical education in rural community hospitals. Through case report-driven medical education at a community hospital, medical students and junior residents learned about the following three themes: perception of medicine, academic career, and responsibility in collaboration. Regarding the perception of medicine, the participants learned about various difficulties in rural family medicine related to VUCA in the medical care of diverse elderly patients in rural contexts. With respect to their academic careers, participants increased their interest in family medicine and realized the importance of case reports in academic careers. They also learned about the realistic implementation of evidence in their cases. Regarding responsibility in collaboration, their responsibilities as researchers increased. Through collaboration, they realized the importance of scheduling to complete case reports, including prompt communication with the co-author.

Case report-driven medical education, based on the results of the perception of medicine, can prepare medical students and residents for the VUCA world that is emerging in medicine. VUCA refers to volatile, uncertain, complex, and ambiguous conditions, describing the present medical conditions across the world [38,39]. After the COVID-19 pandemic, education in the VUCA world has been emphasized in medical education and applied to geriatric care [40]. In this study, medical students and junior residents learned about the concepts of VUCA by writing case reports of complicated older patients at a rural community hospital. Medical students and residents had to consider complicated backgrounds, volatile conditions, clinical course uncertainty, and decision-making ambiguity when writing case reports of the complicated cases that they had not encountered at urban general and university hospitals. Writing the case reports required active involvement from medical students and residents, and provided opportunities to consider VUCA aspects of geriatric care in rural hospitals [41,42]. Family medicine education can also integrate the concept of VUCA in educating challenging situations of the care of elderly patients with multimorbidity [43]. This study is the first to show the effects of CBE using case report writing to learn the concept of VUCA in rural contexts.

An academic career as a medical professional can be beneficial for medical students and residents seeking to enhance their professional careers. Academic research on the compilation of clinical experience with scientific data references begins with writing case reports, which requires systematic consideration of patient conditions [44,45]. In this study, writing case reports allowed participants to realize the importance of recording their experiences and sharing evidence with other medical professionals. Through the discussion of case reports with family medicine residents and teachers, the participants realized the importance of systems thinking in geriatric patient care and patient-centered methods, leading to an interest in family medicine. Furthermore, learners’ implementation of evidence-based medicine (EBM) can become realistic through the experience of writing case reports of complicated older patients. Previous studies have shown that EBM in geriatric care includes VUCA factors and requires shared decision-making (SDM) with patients and their siblings [46,47]. SDM learning can be promoted through concrete cases and discussions with peers [48,49]. As this study shows, writing case reports can provide participants with opportunities to apply EBM with SDM to the uncertainty of geriatric care.

Writing case reports can facilitate various ways of cooperation among medical students, residents, and teachers, enhancing trainees’ responsibility in collaboration. Improved relationships among them could be enhanced through writing case reports. Writing case reports is a type of PBME shown to be an effective way of teaching and learning critical thinking and problem-solving, as exemplified by this study’s other themes [50,51]. Rural family medicine education lacks medical resources compared to urban situations, so family medical teachers can teach critical skills as physicians by writing case reports for medical learners’ cognizance. Furthermore, in case-based learning, the skills and attitudes toward collaboration can be nurtured through various collaborations and discussions with team members [52]. Learners can realize their responsibility as members to accomplish their cases [53]. As this study shows, participants realized their responsibility as researchers. This realization might lead to more importance being placed on scheduling and prompt communication with the co-authors, so as not to delay the progression of writing case reports. This learning is essential for students’ prospective careers as medical doctors and scientists, and preparation for the future [54,55]. In addition, rural family medicine education does not include fewer stakeholders and team members, and such situations are beneficial for establishing their relationship compared to urban contexts. The established relationship between rural medical teachers and learners can facilitate case report-driven medical education effectively. Thus, case report-driven medical education can fit rural CBME regarding family medicine. Overall, writing case reports in PBME can be an effective way to teach and learn in the field of family medicine in the VUCA world. It can help students develop the skills and knowledge of medical scientists for their future careers, and enable them to apply their skills and knowledge in a real-life setting.

This study has several limitations. The first concerns the motivation of the participants regarding learning about family medicine and case reports, because the motivation of learning might affect the learning processes. The participants were motivated to learn about family medicine in community hospitals. The transferability to all medical students and residents may be problematic because not all are interested in writing case reports. In addition, this study was performed at only one rural community hospital. Thus, to overcome this limitation, the researchers clarified the learning content of multiple participants through iterative data collection and comprehensive descriptions of the contexts and learning methods. Another limitation is reliability; to improve reliability, we used iterative data analysis and a long duration of data collection. Future studies should investigate effective educational methods in other regions and in international contexts, including this study’s theory. Additionally, the first author coded the data transcripts, which could have affected the credibility of this study. To combat this, the second researcher reviewed the process of coding, concepts, and themes for investigator triangulation.

## 5. Conclusions

Case report-driven medical education in community hospitals can drive the learning of medical students and junior residents, encourage them to seek academic careers as researchers and family physicians, and help them recognize the importance of responsibility in collaboration. The continual provision of case report-driven medical education can improve the understanding of family medicine in the VUCA world and increase the number of family physicians and academic clinicians in rural contexts.

## Figures and Tables

**Table 1 healthcare-11-02270-t001:** Fifty-one case reports published in Cureus in April 2021–March 2023.

Year	First Author	Patient Age	Patient Sex	Title
2021	Adachi	86	Female	A Case of Cast Nephropathy Found as the Cause of Severe Renal Failure
2021	Yamane	73	Male	Left Lower Abdominal Pain as an Initial Symptom of Multiple Myeloma
2021	Amano	87	Male	Natural Killer T Cell Intravascular Lymphoma with Presentation of Musculoskeletal Pain: A Case Report
2022	Sawa	87	Male	Bilateral Intracardiac Microbubbles in a Patient with Giant Hiatus Hernia: A Case Report
2022	Mouri	76	Female	Peritoneal Cancer Mimicking Sclerosing Mesenteritis: A Case Report
2022	Ikeda	70	Male	The Persistent Approach to Diagnose Infectious Hepatic Cysts in a Patient with Recurrent Fever: A Case Report
2022	Ohta	88	Male	Severe Immune Thrombocytopenic Purpura Following Influenza Vaccination: A Case Report
2022	Nakayama	86	Male	Intercostal Muscle Abscesses in Infective Endocarditis Associated with Migratory Deposition of Calcium Pyrophosphate
2022	Tokonami	82	Female	Pericarditis With Cardiac Tamponade Mimicking Yellow Nail Syndrome in a Patient with Rheumatoid Arthritis and a Paucity of Joint Symptoms
2022	Ohta	86	Male	Fatal Bleeding from a Common Iliac Arterio-Ureteral Fistula in an Older Patient
2022	Ohta	78	Male	Difficulty in Diagnosing and Treating a Prostate Abscess with Bacterial and Fungal Coinfection in an Immunocompromised Patient
2022	Ohta	91	Female	A Rare Case of Herpes Esophagitis in an Immunocompetent Elderly Patient
2022	Ohta	99	Female	Acute Cholecystitis in an Elderly Patient with Antineutrophil Cytoplasmic Antibody-Associated Vasculitis: A Case Report
2022	Yamashita	69	Male	Herpes Simplex Virus Pneumonia Mimicking Legionella Pneumonia in an Elderly Patient with Heart and Liver Failure
2022	Ohta	65	Female	Serotonin Syndrome Triggered by Overuse of Caffeine and Complicated with Neuroleptic Malignant Syndrome: A Case Report
2022	Tachibana	65	Male	A Case of Complicated Pneumonia Caused by *Klebsiella ozaenae*
2022	Kusunoki	83	Male	*Yersinia pseudotuberculosis* Bacteremia Complicated by Rhabdomyolysis
2022	Yamauchi	94	Female	Pseudogout as a Cause of Fever of Unknown Origin Following Staphylococcal Bacteremia in an Older Patient
2022	Watase	86	Male	Hemodialysis-Related Pericarditis with Cardiac Tamponade
2022	Mabuchi	30	Female	Granulomatous Mastitis with Erythema Nodosum During Pregnancy: A Case Report
2022	Fukunaga	18	Female	Adult-Onset Still’s Disease with Severe Hyperferritinemia and the Asian Salmon-Pink Rash: A Case Report
2022	Oshikiri	72	Male	Food Aspiration Induced Hypoxic Encephalopathy Leading to Status Epilepticus
2022	Amano	79	Female	Iliopsoas Pyomyositis with Bacteremia at an Early Stage of Presentation in a Temperate Region
2022	Okuyasu	89	Male	Coexistence of Pancytopenia and Myositis After Developing COVID-19
2022	Ohta	91	Male	Peripheral T-cell Lymphoma with Acute Exacerbating Fatigue and Chest Pain: A Case Report
2022	Ohta	91	Female	Giant Cell Arteritis Mimicking Polymyalgia Rheumatica: A Challenging Diagnosis
2022	Katagiri	69	Male	Hashimoto Encephalopathy of a Middle-Aged Man with Progressive Symptoms of Dementia
2022	Nakano	72	Male	A Case of Legionella Pneumonia in an Older Patient Without Typical Exposure to a Susceptive Environment
2022	Tokonami	82	Male	Autoimmune Vasculitis Causing Acute Bilateral Lower Limb Paralysis
2022	Yoshioka	91	Female	Seronegative Ocular Myasthenia Gravis in an Older Woman with Transient Dizziness and Diplopia
2022	Ohta	91	Female	A Rare Case of Herpes Esophagitis in an Immunocompetent Elderly Patient
2022	Tanaka	87	Female	Device-Related Thrombotic Microangiopathy in an Elderly Patient with a History of Aortic Surgery
2022	Yamamoto	91	Female	Localized Pancreatitis in an Elderly Patient Without Suspected Etiology
2022	Takagi	72	Female	Secondary Failure of Tocilizumab in Treating Elderly-Onset Rheumatoid Arthritis with Systemic Symptoms Complicated by Diverticulum Perforation
2022	Ohta	65	Female	Anti-Mi2 Antibody Positive Dermatomyositis with Hyper-Elevated Creatine Kinase: A Case Report
2022	Nishikura	90	Female	Refractory Immunoglobulin A (IgA) Vasculitis in an Elderly Patient: A Case Report
2022	Okuyasu	34	Male	Hypoglossal Nerve Palsy Following COVID-19 Vaccination in a Young Adult Complicated by Various Medicines
2022	Hayashi	24	Male	Eosinophilic Gastroenteritis in the Small Intestine Mimicking Eosinophilic Granulomatosis with Polyangiitis in a Young Male Patient
2022	Murakami	62	Female	Gallstone Hepatitis Caused by Transient Common Bile Duct Obstruction in a Middle-Aged Woman
2022	Furuta	85	Male	Possible Macrophage Activation Syndrome Caused by Endoscopic Retrograde Cholangiopancreatography for Bacteremia Due to Chronic Cholelithiasis
2022	Aoe	88	Female	Seronegative Rheumatoid Arthritis in an Elderly Patient with Anemia: A Case Report
2022	Nanyoshi	90	Female	Tuberculous Pleurisy Diagnosed from Massive Pleural Effusion in an Older Patient with No History of Tuberculosis
2022	Katsube	86	Female	A Case of Capillary Leak Syndrome and Intestinal Ischemia Caused by Rheumatoid Vasculitis
2023	Horinishi	65	Female	Subacute Bacterial Cellulitis with a Subacute Clinical Course With Difficulty in Distinguishing from Sjögren’s Syndrome: A Case Report
2023	Uchiyama	72	Male	Multiple Prostatic Abscesses Caused by *Staphylococcus aureus* Without Physical Findings in an Immunosuppressed Older Patient
2023	Tabata	35	Female	A Case of Pseudoappendicitis Caused by Campylobacter Enteritis Diagnosed by Gram Staining and Direct Microscopic Investigation of Stool Specimen
2023	Ohta	78	Female	Difficulty in Diagnosing Anti-neutrophil Cytoplasmic Antibody-Related Vasculitis with Interstitial Pneumonia and in Ascertaining the Cause of Associated Hematochezia: A Case Report
2023	Takebuchi	67	Male	Acute Exacerbation of Hypereosinophilic Syndrome Complicated with Dermatitis, Enteritis, and Myositis: A Case Report
2023	Minatogawa	73	Male	Meningitis with *Staphylococcus aureus* Bacteremia in an Older Patient with Nonspecific Symptoms: A Case Report
2023	Akashi	86	Female	Deciding a Treatment Plan for an Older Patient with Severe Idiopathic Pulmonary Fibrosis: A Case Report
2023	Naito	72	Female	Right-Sided Urinary Extravasation Caused by a Ureteral Stone and Associated with Peritonitis in an Older Woman

**Table 2 healthcare-11-02270-t002:** Results of the thematic analysis approach.

Theme	Concepts
Perception toward medicine	Volatility in geriatric care
	Uncertainty of clinical course
	Complexity of medicine
	Ambiguity of patient’s decisions
Academic career	Interest in family medicine
	Importance of case reports in an academic career
	Realistic implementation of evidence
Responsibility in collaboration	Responsibility as a researcher
	Importance of scheduling
	Prompt communication with the co-author

## Data Availability

The datasets used and/or analyzed during the current study may be obtained from the corresponding author upon reasonable request.

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
