# Peer review of "Case Report-Driven Medical Education in Rural Family Medicine Education: A Thematic Analysis"

_healthcare, 2023, doi:10.3390/healthcare11162270_

Round 1

Reviewer 1 Report (Previous Reviewer 1)

The authors have taken into consideration the previous comments. They have addressed the issues around explaining the methodology and the reasoning behind their analytical approach so the paper now concisely introduces the grounded theory approach and justifies the methodology and how the analysis was conducted with appropriate academic rigour.

I hope to see this in publication and that it will encourage further use of this methodology in medical education research.

Author Response

The authors have taken into consideration the previous comments. They have addressed the issues around explaining the methodology and the reasoning behind their analytical approach so the paper now concisely introduces the grounded theory approach and justifies the methodology and how the analysis was conducted with appropriate academic rigour.

I hope to see this in publication and that it will encourage further use of this methodology in medical education research.

Response

Thank you for your feedback.

Reviewer 2 Report (New Reviewer)

Thanks for the opportunity to review this paper concerning base based education in a rural hospital in Japan. This is an interesting research area. I can see the authors have already addressed another reviewer's concerns around CBE and the changes have strengthened the paper considerably.

I am concerned that the authors have not demonstrated a strong methodology. The word 'ethnographic' is used in the abstract yet there is no mention of ethnography in the methods. The approach is grounded theory, yet the research questions are not aligned to grounded theory design (e.g. what governs social processes? why do people do what they do?), participants were only interviewed once, and it seems thematic analysis was undertaken rather than grounded theory. The findings are reported exactly as I would expect for the thematic analysis. There is no theory presented.

The authors are not alone. I frequently see authors reporting findings that they state are grounded theory but in fact the evidence does not strongly support grounded theory. If you were to publish this in its current form you might come under scrutiny. To avoid that as I see it, you have two options: either strengthen your evidence of grounded theory, remove the term ethnographic, and provide a theory   OR   state you have undertaken general purpose qualitative research with thematic analysis, and remove the term axial coding from methods.

minor: Abstract: CBME acronym is used yet this is not written in full. 

The conclusions are sound and will be of international interest to health professions educators.

Author Response

Thanks for the opportunity to review this paper concerning base based education in a rural hospital in Japan. This is an interesting research area. I can see the authors have already addressed another reviewer's concerns around CBE and the changes have strengthened the paper considerably.

I am concerned that the authors have not demonstrated a strong methodology. The word 'ethnographic' is used in the abstract yet there is no mention of ethnography in the methods. The approach is grounded theory, yet the research questions are not aligned to grounded theory design (e.g. what governs social processes? why do people do what they do?), participants were only interviewed once, and it seems thematic analysis was undertaken rather than grounded theory. The findings are reported exactly as I would expect for the thematic analysis. There is no theory presented. The authors are not alone. I frequently see authors reporting findings that they state are grounded theory but in fact the evidence does not strongly support grounded theory. If you were to publish this in its current form you might come under scrutiny. To avoid that as I see it, you have two options: either strengthen your evidence of grounded theory, remove the term ethnographic, and provide a theory   OR   state you have undertaken general purpose qualitative research with thematic analysis, and remove the term axial coding from methods.

Response

Thank you for your feedback. We agree with your suggestion. We have revised the abstract description accordingly as follows:

“Using a qualitative design based on a thematic analysis approach, data were collected through semi-structured interviews..” (Lines 15–16)

Additionally, based on your suggestion, we have revised the methods of the research to incorporate thematic analysis throughout the research.

minor: Abstract: CBME acronym is used yet this is not written in full. 

Response

Thank you for pointing this out. We have spelled out CBME in the Abstract as “community-based medical education.”

The conclusions are sound and will be of international interest to health professions educators.

Response

Thank you for your feedback.

Round 2

Reviewer 2 Report (New Reviewer)

I think the authors have made a good decision to revise this as thematic analysis. I note page 4 (line 188) still refers to 'the theory', although you have not created a theory. So I'd recommend removing that. Otherwise its looking good, thanks.

Author Response

Response

Thank you for the comment. Based on the suggestion, we have deleted the suggested part as follows.

“Finally, concepts and themes were discussed by both researchers, who ultimately agreed on the final themes.” (Line 188 to 189)

This manuscript is a resubmission of an earlier submission. The following is a list of the peer review reports and author responses from that submission.

Round 1

Reviewer 1 Report

Very interesting to see a grounded theory approach applied to a medical context. The study is well-designed and thoroughly described. The research question of what and how students learn is successfully answered. I found the discussion section enlightening regarding the VUCA concept. I congratulate the authors on a clear, informative presentation of well-conducted and interesting research.

My additional specific comments would be: This is a very topical area that will be of interest to many readers. While the focus is on a rural Japanese context, parallels can be drawn with other rural contexts. By adopting a grounded theory approach, the authors can set the specifics of the study context and engage in the sort of in-depth analysis that allows for some interesting conclusions to be drawn that are much more broadly applicable. This is a real strength of the chosen methodology and also allows for a more immediate action research approach so not requiring comparisons or control groups. The paper would probably help any readers unfamiliar with the methodology and more qualitative approaches by including some standard references to the approach and to the thematic analysis methods used (e.g. Watling CJ, Lingard L. Grounded theory in medical education research: AMEE Guide No. 70. Med Teach. 2012;34(10):850-61. doi: 10.3109/0142159X.2012.704439. Epub 2012 Aug 22. PMID: 22913519. Braun, V. & Clarke, V. (2006). Using thematic analysis in psychology. Qualitative research in psychology, 3 (2), 77-101.)

Some minor errors which can easily be picked up with another proofread. eg line 121

Author Response

Very interesting to see a grounded theory approach applied to a medical context. The study is well-designed and thoroughly described. The research question of what and how students learn is successfully answered. I found the discussion section enlightening regarding the VUCA concept. I congratulate the authors on a clear, informative presentation of well-conducted and interesting research.

My additional specific comments would be: This is a very topical area that will be of interest to many readers. While the focus is on a rural Japanese context, parallels can be drawn with other rural contexts. By adopting a grounded theory approach, the authors can set the specifics of the study context and engage in the sort of in-depth analysis that allows for some interesting conclusions to be drawn that are much more broadly applicable. This is a real strength of the chosen methodology and also allows for a more immediate action research approach so not requiring comparisons or control groups. The paper would probably help any readers unfamiliar with the methodology and more qualitative approaches by including some standard references to the approach and to the thematic analysis methods used (e.g., Watling CJ, Lingard L. Grounded theory in medical education research: AMEE Guide No. 70. Med Teach. 2012;34(10):850-61. doi: 10.3109/0142159X.2012.704439. Epub 2012 Aug 22. PMID: 22913519. Braun, V. & Clarke, V. (2006). Using thematic analysis in psychology. Qualitative research in psychology, 3 (2), 77-101.) 

Response:                                                                                                                             

Thank you for your constructive response. We agree with the feedback, and have revised the analysis part, including how to analyze concretely, why VOCA was used in the analysis and results, and referring to suggested articles as follows.

“The grounded theory approach was used to clarify the participants’ learning through case report-driven medical education [36]. All the field notes and interviews were recorded and transcribed verbatim. After reviewing the field notes and conducting the first five semi-structured interviews, the first researcher, R.O., coded the data and developed codebooks based on the repeated reading of field notes as initial coding for reliability. This study used process and concept coding methods [37]. The coded contents in the initial coding were shared with the second researcher, C.S., and the two researchers discussed the contents for triangulation. Then, R.O. continued initial coding individually, creating tentative concepts. The interview contents were analyzed iteratively during the research period after finishing each participant’s case report learning for theoretical saturation. After the initial coding and creating tentative concepts, R.O. induced, merged, deleted, and refined the tentative concepts and created themes by going back and forth between the research materials and initial coding for axial coding. During the code refinement process, axial coding was used to elaborate on tentative concepts and themes. R.O. and C.S. discussed the concepts and themes created through axial coding for further triangulation. Through the discussion, the concept of VUCA (volatility, uncertainty, complexity, ambiguity) appeared, which was incorporated into the themes and concepts regarding the previous article on the present medical education [38]. Finally, the theory, comprising concepts and themes, was discussed by both researchers, who ultimately agreed on the final themes.” (Lines 164–183).

Reviewer 2 Report

Dear Authors

Thank you for the opportunity to review your manuscript.

The manuscript deals with a topical issue and has the potential to be of interest to an international audience. However, there are significant flaws, especially related to the methodology and the presentation of results, that must be addressed. I address some of my concerns below: 

Background

-          Some references relating to PBME seem inadequate. References 1-3 relate to project-learning more broadly. I would expect the authors to refer to medical education academic journals when discussing PBME. Alternatively, the authors should refer to project-based learning (PBL) in the context of medical education, rather than PBME.

-          What do the authors mean by ‘PBME can be performed in various situations’? (line 30).

-          Reference 17 relates to social work, not medical education.

-          It is unclear how CA and LPP relate to case report writing in the context of PBME.

-          The last paragraph is very long and detracts from legibility. The authors should consider introducing the issues relating to rural contexts in a separate paragraph (approximately line 66).

-          In line 68, do the authors mean that case-driven has not been explored adequately in rural medical education settings? Case-based learning has been explored in medical education more broadly (see Scaffolding medical student knowledge and skills: team-based learning (TBL) and case-based learning (CBL) by Burgess et al, 2021, for example).

Methods

Section 2.1 – Setting

-          Grammar (tense) issues in line 79.

-          What is the difference between ‘residents’ and ‘tertiary hospital residents’? (line 81)

-          What are the ‘clinical situations’ the authors refer to in line 82? Are they the same as the ‘clinical settings’ referred in lines 83-84?

-          Do medical students and junior residents train together in Japan? More detail and clarity is needed on the Japanese medical education context – the information provided in lines 84-90 is difficult to understand for an international audience not familiar with the Japanese medical education curriculum.

Section 2.2 – Participants

-          What do the authors mean by ‘were assigned to participate in the CBME’? Does it mean that not all students participate in it as part of their curriculum?

-          How were participants recruited?

-          The authors’ description of the data collection process in lines 98-101 is unclear. Was the data collected through observation, field notes and interviews or only through interviews?

Section 2.3 – Care report-driven medical education

-          The description of the case report-driven medical education in section 2.3 is too long. Some of the detail is unnecessary (for example, the definition in lines 105-108). This section should be succinct and clear. What is meant by ‘good or bad conditions’?

-          What is the meaning of lines 134-135? Who are the ‘researchers’ and what ‘international medical journals’ were these case reports submitted to?

Section 2.4 – Measurements

-          Section 2.4 is entitled ‘measurements’. This should be ‘data collection’ or something similar, as ‘measurements’ seems to imply a quantitative research paradigm.

-          What are the ‘reflection sheets’ mentioned in line 150? More detail is needed.

-          What is the meaning of the acronym ‘CRDE’ in line 151?

Section 2.5 – Analysis

-          What are the ‘process and concept coding methods’ referred to in line 158?

-          Could the authors clarify their data analysis process by providing examples of how ‘concepts’, ‘themes’ and the ‘theory’ were arrived at?

-          What is the ‘theory’ that was arrived at and that was discussed by ‘both researchers’ as the authors state in line 164?

-          Were the interviews recorded? Were they transcribed?

-          How were the field notes and reflection sheets used in the analysis?

-          The description of how theoretical saturation was achieved in lines 162-164 is unclear.

Section 2.6 – Ethical consideration

-          What are the ‘questionnaires’ and ‘conferences’? Are the ‘conferences’ the interviews that took place in the conference room? Did participants complete a questionnaire?

Results

Section 3.1 – Demographic data

-          There is a major flaw with the presentation of the demographic data in the manuscript and in Table 1. The aim of the research is to explore students’ experience of case report-driven PBME, thus, the demographic data relevant to this paper is students’ and residents’ data, not patients’ data. Readers need to know about participants in the research, that is, students and residents.

Section 3.2 – Results of the ground theory approach

-          As indicated previously, it is unclear how the themes and ‘concepts’ were arrived at.

-          The authors state in line 154 that ‘an inductive grounded theory approach’ was used. Yet, the ‘concepts’ identified under the first theme are aligned with the notion of VUCA, thus implying a deductive rather than inductive approach. The authors’ description of the overall methodology as ‘grounded theory’ is problematic and does not seem to fit the data analysis method demonstrated in the presentation of results.

-          The term VUCA is introduced for the first time in this section. Given that this notion is heavily relied upon in the discussion, it should be introduced in the Background section. I note that VUCA was coined within the context of organisational management. Its application to medical education needs to be discussed earlier in the paper (Background section).

-          Generally, the section lacks depth and rigour, with no demonstration of level of agreement/disagreement within the sample (discordant voices/contrasting experiences or opinions), no exploration of the impact of seniority, i.e., do the experiences of students vary significantly compared with that of Junior Residents, and, critically, no detailed exploration of the impact of the rural setting other than general references to differences between community hospitals and university and tertiary hospitals. This is a major flaw, given that the authors state in lines 70-71 that the research question was: “What and how do medical students and residents learn through case report-driven PBME in rural contexts?”

-          It is unclear whether participants’ experiences of case report-driven PBME were influenced by the rural setting or, rather, by their patients’ age. For example, under the first ‘theme’, the references to VUCA seem to be all related to treating ‘elderly patients’.

-          There is no reference to the impact of COVID-19 in the results section; this is surprising given that the research took place during the pandemic and some patients had conditions associated with COVID-19.

Discussion

-          As noted earlier, the authors rely heavily on the notion of VUCA in their discussion. This notion must be introduced in the Background.

-          What do the authors mean by ‘Family medicine education can also involve the care of elderly patients with multimorbidity from the perspective of VUCA’?

-          What is meant by ‘Furthermore, the implementation of evidence-based medicine (EBM) can be made realistic’?

-          Generally, the discussion fails to explore the role played by the rural context. How are students’ experiences different to those of their urban counterparts? What is it about the rural context that influences student learning? Is it having fewer resources? (as implied in the Background section) Is it the patient profile? (more elderly patients?)

-          Student motivation affects learning on all subject matters. Do the authors think that student motivation might have introduced bias in this study? Or is the bias introduced by some students' willingness to participate in the study? The meaning of lines 390-391 is unclear.

-          Transferability and reliability are two different concepts.

-          What do the authors mean by ‘theoretical triangulation’?

Dear Authors

The paper would benefit from editing as the authors' meaning is not always clear. See examples in my comments above.

Author Response

Dear Authors

Thank you for the opportunity to review your manuscript.

The manuscript deals with a topical issue and has the potential to be of interest to an international audience. However, there are significant flaws, especially related to the methodology and the presentation of results, that must be addressed. I address some of my concerns below: 

Background

-          Some references relating to PBME seem inadequate. References 1-3 relate to project-learning more broadly. I expect the authors to refer to academic medical education journals when discussing PBME. Alternatively, the authors should refer to project-based learning (PBL) in the context of medical education rather than PBME.

Response:

Thank you for your illuminating feedback and positive remarks, here and below. We agree with your comments, and have revised the references by including research about PBL in medical education as follows:

  1. Keator, C.S.;D. Vandre; A.M. Morris, The Challenges of Developing a Project-Based Self-

Directed Learning Component for Undergraduate Medical Education. Med Sci Educ. 2016, 26, 4, 801–805.

  1. Stentoft, D. Problem-based projects in medical education: extending PBL practices and broadening learning perspectives. Adv Health Sci Educ Theory Pract. 2019, 24, 5, 959–
  2. Kim, K.J. Project-based learning approach to increase medical student empathy. Med Educ Online. 2020, 25, 1742965. DOI:10.1080/10872981.2020.1742965.

-          What do the authors mean by ‘PBME can be performed in various situations’? (line 30).

Response:

We revised the part in question as follows:

“Project-based medical education (PBME) is a teaching method in which students work on real-world projects to learn and apply concepts and skills they have been taught [1]. PBME can be performed in undergraduate and postgraduate medical education [2,3].” (Lines 28–30).

-          Reference 17 relates to social work, not medical education.

Response:

We agree with your remark, and revised the reference regarding patient encounter issues in medical education as follows:

Murtagh, G.M., A critical look at ideas, concerns and expectations in clinical communication. Med Educ. 2023, 57, 4, 331–336.

-          It is unclear how CA and LPP relate to case report writing in the context of PBME.

Response:

We revised the paragraph regarding the theoretical framework of case report-driven medical education respecting CA and LPP as follows:

“Cognitive apprenticeship (CA) and legitimate peripheral participation (LPP) effectively involve medical trainees [24,25]. These frameworks allow trainees to gradually increase the extent of their participation in clinical medicine [24,25]. Writing case reports with supervisors or medical teachers and examining patients together, may also motivate trainees to participate in clinical medicine [22]. Sequential methods of writing case reports, accompanied with gradual exposure to medical examination of patients, are important for case report-driven medical education in respect of CA and LPP [24–26]. Writing case reports may also allow supervisors and medical teachers to reflect on patient care [27].” (Lines 59–67).

-          The last paragraph is very long and detracts from legibility. The authors should consider introducing the issues relating to rural contexts in a separate paragraph (approximately line 66).

Response:

We concur, and revised the last paragraph by dividing it into two paragraphs, theoretical background of case report driven medical education, and research question:

“Case report-driven medical education within the PBME framework can effectively drive student and resident learning, andincrease their involvement in real clinical practice. In medical education, the involvement of medical trainees in clinical situations can be integral for better learning. Cognitive apprenticeship (CA) and legitimate peripheral participation (LPP) effectively involve medical trainees [24,25]. These frameworks allow trainees to gradually increase the extent of their participation in clinical medicine [24,25]. Writing case reports with supervisors or medical teachers and examining patients together, may also motivate trainees to participate in clinical medicine [22]. Sequential methods of writing case reports, accompanied with gradual exposure to medical examination of patients, are important for case report-driven medical education in respect of CA and LPP [24–26]. Writing case reports may also allow supervisors and medical teachers to reflect on patient care [27]. Case reports do not require many resources, and can be completed through discussions among trainees, supervisors, and teachers [27].

Case report-driven medical education is particularly useful in resource-limited medical education, such as rural community-based medical education (CBME), which suffers from a lack of medical teachers and teaching resources [28–30]. In addition, learning from writing case reports of complicated older patients in rural contexts could enhance understanding of varieties of older patients’ presentations such as VUCA (volatility, uncertainty, complexity, and ambiguity), describing challenging situations in caring for older people. However, case report-driven medical education has not been investigated adequately in contemporary medical education. Therefore, we posed the research question: “What do medical students and residents learn about patients’ care and physician’s academic careers through case report-driven PBME in rural community hospitals?” Clarification of specific case report-driven medical education methods and learning outcomes in resource-limited settings can motivate medical educators in rural contexts to adopt PBME and case reports, thereby benefiting rural CBME. Therefore, this study aimed to clarify the specific methods and learning outcomes of case report-driven medical education in rural CBME.” (Lines 58–83).

-          In line 68, do the authors mean that case-driven has not been explored adequately in rural medical education settings? Case-based learning has been explored in medical education more broadly (see Scaffolding medical student knowledge and skills: team-based learning (TBL) and case-based learning (CBL) by Burgess et al, 2021, for example).

Response:

We agree with the feedback. As you stated, case-based learning was investigated by many researchers. Our focus was not case-based, but case report-driven medical education in rural contexts.

Methods

Section 2.1 – Setting

-          Grammar (tense) issues in line 79.

Response:

We revised the tense in the paragraph.

-          What is the difference between ‘residents’ and ‘tertiary hospital residents’? (line 81)

Response:

We noted the valid question, and revised the sentence as follows:

“The hospital provides CBME to medical students and residents of medical universities and tertiary hospitals.” (Lines 88–89).

-          What are the ‘clinical situations’ the authors refer to in line 82? Are they the same as the ‘clinical settings’ referred in lines 83-84?

Response:

We deleted the second sentence to remove the ambiguity.

-          Do medical students and junior residents train together in Japan? More detail and clarity is needed on the Japanese medical education context – the information provided in lines 84-90 is difficult to understand for an international audience not familiar with the Japanese medical education curriculum.

Response:

We added the explanation about medical students and residents’ allowed behaviors in the hospital as follows:

“The range of allowed practice of medical students and residents differed: medical students needed the observation of medical teachers in each encounter with patients; but residents could examine patients individually, and then had to discuss managingthem with medical teachers before prescribing and ordering tests and specific procedures.” (Lines 96–100).

Section 2.2 – Participants

-          What do the authors mean by ‘were assigned to participate in the CBME’? Does it mean that not all students participate in it as part of their curriculum?

We revised the sentence as follows:

“Between April 2021 and March 2023, 53 medical students and 16 junior residents participated voluntarily in the CBME curriculum, which included family medicine at Unnan City Hospital.” (Lines 105–107).

-          How were participants recruited?

-          The authors’ description of the data collection process in lines 98-101 is unclear. Was the data collected through observation, field notes and interviews or only through interviews?

Response:

We revised section 2.2 comprehensively, including the references to participant inclusion and data collection, as follows:

“CBME education at Unnan City Hospital accepted medical students and residents from anywhere in Japan. Between April 2021 and March 2023, 53 medical students and 16 junior residents participated voluntarily in the CBME curriculum, which included family medicine at Unnan City Hospital. The training aimed to produce competencies in general medicine areas required by Japan, such as person-centered care, comprehensive and integrative approaches, interprofessional work, community orientation, professionalism, and systematic practice [34,35]. Ultimately, 45 medical students, seven junior residents, and eight family medicine residents agreed with writing case reports during the training, and participating in this study. We used purposive sampling to address the research purposes of ethnography through observation, field notes, and interviews. Data were collected using field notes and semi-structured interviews to investigate the concrete experience and learning of the participants.” (Lines 104–115).

Section 2.3 – Care report-driven medical education

-          The description of the case report-driven medical education in section 2.3 is too long. Some of the detail is unnecessary (for example, the definition in lines 105-108). This section should be succinct and clear. What is meant by ‘good or bad conditions’?

-          What is the meaning of lines 134-135? Who are the ‘researchers’ and what ‘international medical journals’ were these case reports submitted to?

Response:

We revised the explanatory description about case report-driven medical education as follows:

“Case report-driven medical education began in the first week of rural CBME training, during which a family medicine teacher explained case report-driven medical education. 

Medical students examined cases with family medicine residents and educators, taking clinical histories and noting physical examinations. Based on these clinical notes, they discussed the patients’ clinical conditions. In some cases, medical educators provided several articles and learning materials to medical students and junior residents to facilitate their understanding of patients’ conditions. Based on the discussions and relevant reading material, medical students and residents progressed in writing their case reports. Cases of patients actively infected with COVID-19 were not allocated to participants of this study because of the risk of infections and the limitation of interaction with patients.

The case reports comprised four sections: introduction, case presentation, discussion, and conclusions. The writing was facilitated by discussions among medical students, junior residents, family medicine residents, and teachers. Case selection was performed through discussions with the participants. First, medical students and junior residents summarized patients’ diseases and specific features. Residents and teachers of family medicine reviewed and revised these descriptions. Second, the medical students and junior residents described the patients’ clinical histories, physical examinations, specific tests, and treatments. When the medical students and junior residents struggled with their descriptions, they discussed them with family medicine residents and teachers. They re-examined the patients with family medicine residents or medical teachers to supplement their documentation.

Third, the medical students and residents described the discussion and conclusions based on what they had learned. Through discussions with family medicine residents and teachers, they selected topics and started to draft their discussion sections. Family medicine residents and teachers reviewed the medical students’ descriptions, and revised the content regarding paragraphs and academic writing. All documentation for the case reports was provided by medical students, junior residents, family medicine residents, and medical teachers. Finally, the family medicine residents and medical teachers reviewed and revised the content of the case reports and submitted each manuscript to an international medical journal, Cureus.” (Lines 117–146).

Section 2.4 – Measurements

-          Section 2.4 is entitled ‘measurements’. This should be ‘data collection’ or something similar, as ‘measurements’ seems to imply a quantitative research paradigm.

Response:

We changed the word to data collection.

-          What are the ‘reflection sheets’ mentioned in line 150? More detail is needed.

Response:

We deleted the phrase as it was redundant.

-          What is the meaning of the acronym ‘CRDE’ in line 151?

Response:

We deleted the phrase as it was redundant.

Section 2.5 – Analysis

-          What are the ‘process and concept coding methods’ referred to in line 158?

-          Could the authors clarify their data analysis process by providing examples of how ‘concepts’, ‘themes’ and the ‘theory’ were arrived at?

-          What is the ‘theory’ that was arrived at and that was discussed by ‘both researchers’ as the authors state in line 164?

-          How were the field notes and reflection sheets used in the analysis?

-          The description of how theoretical saturation was achieved in lines 162-164 is unclear.

Response:

We revised the analysis part by adding to the concrete process of analysis and including concepts, themes and theory building through the researchers’ collaboration and discussion as follows:

“The grounded theory approach was used to clarify the participants’ learning through case report-driven medical education [36]. All the field notes and interviews were recorded and transcribed verbatim. After reviewing the field notes and conducting the first five semi-structured interviews, the first researcher, R.O., coded the data and developed codebooks based on the repeated reading of field notes as initial coding for reliability. This study used process and concept coding methods [37]. The coded contents in the initial coding were shared with the second researcher, C.S., and the two researchers discussed the contents for triangulation. Then, R.O. continued initial coding individually, creating tentative concepts. The interview contents were analyzed iteratively during the research period after finishing each participant’s case report learning for theoretical saturation. After the initial coding and creating tentative concepts, R.O. induced, merged, deleted, and refined the tentative concepts and created themes by going back and forth between the research materials and initial coding for axial coding. During the code refinement process, axial coding was used to elaborate on tentative concepts and themes. R.O. and C.S. discussed the concepts and themes created through axial coding for further triangulation. Through the discussion, the concept of VUCA (volatility, uncertainty, complexity, ambiguity) appeared, which was incorporated into the themes and concepts regarding the previous article on the present medical education [38]. Finally, the theory, comprising concepts and themes, was discussed by both researchers, who ultimately agreed on the final themes.” (Lines 164–183).

-          Were the interviews recorded? Were they transcribed?

Response:

We added an explanation of the data collection.

Section 2.6 – Ethical consideration

-          What are the ‘questionnaires’ and ‘conferences’? Are the ‘conferences’ the interviews that took place in the conference room? Did participants complete a questionnaire?

Response:

We revised the suggested part as follows:

“Participants’ anonymity and confidentiality were ensured throughout the study. All participants completed informed consent forms in writing before participating in this research. All procedures in this study were performed in compliance with the Declaration of Helsinki and its amendments. The Unnan City Hospital Clinical Ethics Committee approved the study protocol (No. 20210032).” (Lines 185–189).

Results

Section 3.1 – Demographic data

-          There is a major flaw with the presentation of the demographic data in the manuscript and in Table 1. The aim of the research is to explore students’ experience of case report-driven PBME, thus, the demographic data relevant to this paper is students’ and residents’ data, not patients’ data. Readers need to know about participants in the research, that is, students and residents.

Response:

We revised the description of demograhic data, including participants’ backgrounds, as follows”

“A total of 45 medical students (six 4th-grade, 16 5th-grade and 23 6th-grade participants), seven junior residents, and eight family medicine residents participated in this research. The female percentage was 43.3% (26/60). Finally, fifty-one case reports were completed and published in the international medical journal, Cureus, from April 2021 to March 2023 (Table 1). The described patients’ average age was 74.5 years old (standard deviation = 18.1), and 47.1 % were male.” (Lines 192–197).

Section 3.2 – Results of the ground theory approach

-          As indicated previously, it is unclear how the themes and ‘concepts’ were arrived at.

Response:

We revised the analysis section by including the concrete analysis process and explaining how the concepts and themes had been arrived at.

-          The authors state in line 154 that ‘an inductive grounded theory approach’ was used. Yet, the ‘concepts’ identified under the first theme are aligned with the notion of VUCA, thus implying a deductive rather than inductive approach. The authors’ description of the overall methodology as ‘grounded theory’ is problematic and does not seem to fit the data analysis method demonstrated in the presentation of results.

-          The term VUCA is introduced for the first time in this section. Given that this notion is heavily relied upon in the discussion, it should be introduced in the Background section. I note that VUCA was coined within the context of organisational management. Its application to medical education needs to be discussed earlier in the paper (Background section).

Response:

We revised the description of grounded theory approaches as reflected below. Regarding the VUCA concepts, we described the process and the concept in the analysis section (Lines 179–181).

“The grounded theory approach was used to clarify the participants’ learning through case report-driven medical education [36]. All the field notes and interviews were recorded and transcribed verbatim. After reviewing the field notes and conducting the first five semi-structured interviews, the first researcher, R.O., coded the data and developed codebooks based on the repeated reading of field notes as initial coding for reliability. This study used process and concept coding methods [37]. The coded contents in the initial coding were shared with the second researcher, C.S., and the two researchers discussed the contents for triangulation. Then, R.O. continued initial coding individually, creating tentative concepts. The interview contents were analyzed iteratively during the research period after finishing each participant’s case report learning for theoretical saturation. After the initial coding and creating tentative concepts, R.O. induced, merged, deleted, and refined the tentative concepts and created themes by going back and forth between the research materials and initial coding for axial coding. During the code refinement process, axial coding was used to elaborate on tentative concepts and themes. R.O. and C.S. discussed the concepts and themes created through axial coding for further triangulation. Through the discussion, the concept of VUCA (volatility, uncertainty, complexity, ambiguity) appeared, which was incorporated into the themes and concepts regarding the previous article on the present medical education [38]. Finally, the theory, comprising concepts and themes, was discussed by both researchers, who ultimately agreed on the final themes.” (Lines 164–183).

-          Generally, the section lacks depth and rigour, with no demonstration of level of agreement/disagreement within the sample (discordant voices/contrasting experiences or opinions), no exploration of the impact of seniority, i.e., do the experiences of students vary significantly compared with that of Junior Residents, and, critically, no detailed exploration of the impact of the rural setting other than general references to differences between community hospitals and university and tertiary hospitals. This is a major flaw, given that the authors state in lines 70-71 that the research question was: “What and how do medical students and residents learn through case report-driven PBME in rural contexts?”

Response

We amended the research question and last paragraph as follows:

“Case report-driven medical education is particularly useful in resource-limited medical education, such as rural community-based medical education (CBME), which suffers from a lack of medical teachers and teaching resources [28–30]. In addition, learning from writing case reports of complicated older patients in rural contexts could enhance understanding of varieties of older patients’ presentations such as VUCA (volatility, uncertainty, complexity, and ambiguity), describing challenging situations in caring for older people. However, case report-driven medical education has not been investigated adequately in contemporary medical education. Therefore, we posed the research question: “What do medical students and residents learn about patients’ care and physician’s academic careers through case report-driven PBME in rural community hospitals?” Clarification of specific case report-driven medical education methods and learning outcomes in resource-limited settings can motivate medical educators in rural contexts to adopt PBME and case reports, thereby benefiting rural CBME. Therefore, this study aimed to clarify the specific methods and learning outcomes of case report-driven medical education in rural CBME.” (Lines 69–83).

-          It is unclear whether participants’ experiences of case report-driven PBME were influenced by the rural setting or, rather, by their patients’ age. For example, under the first ‘theme’, the references to VUCA seem to be all related to treating ‘elderly patients’.

Response:

We agree with the feedback and revised the description of the results regarding the differences between rural community hospitals and tertial and university hospitals as follows:

“While managing and writing about the cases, the participants observed an acute exacerbation of the conditions of elderly patients they did not encounter in their previous learning situations. The participants had experienced younger patients’ medical care before attending the community hospital. One of the junior residents stated, “I have never experienced such older patients with various diseases. They were vulnerable to subtle changes in their circumstances, causing infections and delirium.” (Junior resident 3). The participants realized that elderly patients differed from younger generations and needed special care. One of the residents stated, “Elderly patients can suddenly worsen during their medical care. They respond to standard care, but their renal and liver functions can change with subtle changes in homeostasis. The same treatment may not be easily applied to different patients. Family physicians frequently had to observe the symptoms of elderly patients as opposed to less frequently with younger patients.” (Junior resident 5). The participants observed a difference between the elderly and younger generations regarding their reaction to treatments. By writing case reports, they deepened their understanding of the specialty of geriatric care.” (Lines 219–233).

-          There is no reference to the impact of COVID-19 in the results section; this is surprising given that the research took place during the pandemic and some patients had conditions associated with COVID-19.

Response:

We added the explanation for not using cases infected with COVID-19 as follows:

“Cases of patients actively infected with COVID-19 were not allocated to participants of this study because of the risk of infections and the limitation of interaction with patients.” (Lines 124–126).

Discussion

-          As noted earlier, the authors rely heavily on the notion of VUCA in their discussion. This notion must be introduced in the Background.

Response:

We revised the introduction to include VUCA, as follows:

“Case report-driven medical education is particularly useful in resource-limited medical education, such as rural community-based medical education (CBME), which suffers from a lack of medical teachers and teaching resources [28–30]. In addition, learning from writing case reports of complicated older patients in rural contexts could enhance understanding of varieties of older patients’ presentations such as VUCA (volatility, uncertainty, complexity, and ambiguity), describing challenging situations in caring for older people. However, case report-driven medical education has not been investigated adequately in contemporary medical education. Therefore, we posed the research question: “What do medical students and residents learn about patients’ care and physician’s academic careers through case report-driven PBME in rural community hospitals?” Clarification of specific case report-driven medical education methods and learning outcomes in resource-limited settings can motivate medical educators in rural contexts to adopt PBME and case reports, thereby benefiting rural CBME. Therefore, this study aimed to clarify the specific methods and learning outcomes of case report-driven medical education in rural CBME.” (Lines 69–83).

-          What do the authors mean by ‘Family medicine education can also involve the care of elderly patients with multimorbidity from the perspective of VUCA’?

Response:

We changed the sentence to read as follows:

“Family medicine education can also use the concept of VUCA in educating challenging situations of the care of elderly patients with multimorbidity.” (Lines 388–390).

-          What is meant by ‘Furthermore, the implementation of evidence-based medicine (EBM) can be made realistic’?

Response:

We revised the sentence as follows:

“Furthermore, learners’ implementation of evidence-based medicine (EBM) can become realistic through the experience of writing case reports of complicated older patients.” (Line 400–402).

-          Generally, the discussion fails to explore the role played by the rural context. How are students’ experiences different to those of their urban counterparts? What is it about the rural context that influences student learning? Is it having fewer resources? (as implied in the Background section) Is it the patient profile? (more elderly patients?)

Response:

We revised the discussion part comprehensively. It now includes benefits for case report-driven medical education in rural contexts:

“Case report-driven medical education, based on the results of the perception of medicine, can prepare medical students and residents for the VUCA world that is emerging in medicine. VUCA refers to volatile, uncertain, complex, and ambiguous conditions, expressing the present medical conditions across the world [38,39]. After the COVID-19 pandemic, education in the VUCA world has been emphasized in medical education, and applied to geriatric care [40]. In this study, medical students and junior residents learned the concepts of VUCA by writing case reports of complicated older patients in a rural community hospital. Medical students and residents had to consider complicated backgrounds, volatile conditions, clinical course uncertainty, and decision-making ambiguity when writing case reports of the complicated cases that they had not encountered in the urban general and university hospitals. Writing the case reports required active involvement from medical students and residents, and provided opportunities to consider VUCA aspects of geriatric care in rural hospitals [41,42]. Family medicine education can also use the concept of VUCA in educating challenging situations of the care of elderly patients with multimorbidity [43]. This study is the first to show the effect of project-based medical education using case report writing to learn the concept of VUCA in rural contexts.

An academic career as a medical professional can be beneficial for medical students and residents seeking to enhance their professional careers. Academic research on the compilation of clinical experience with scientific data references begins with writing case reports, which requires systematic consideration of patient conditions [44,45]. In this study, writing case reports allowed participants to realize the importance of compiling their experiences and sharing evidence with other medical professionals. Through the discussion of case reports with family medicine residents and teachers, the participants realized the importance of systems thinking in geriatric patient care and patient-centered methods, leading to an interest in family medicine. Furthermore, learners’ implementation of evidence-based medicine (EBM) can become realistic through the experience of writing case reports of complicated older patients. Previous studies have shown that EBM in geriatric care includes VUCA factors and requires shared decision-making (SDM) with patients and their siblings [46,47]. SDM learning can be promoted through concrete cases and discussions with peers [48,49]. As this study shows, writing case reports can provide participants with opportunities to apply EBM with SDM to the uncertainty of geriatric care.

Writing case reports can facilitate various ways of cooperation among medical students, residents, and teachers, enhancing trainees’ responsibility in collaboration. Improved relationships among them could be enhanced through writing case reports. Writing case reports is a type of PBME shown to be an effective way to teach and learn critical thinking and problem-solving, as exemplified by this study’s other themes [50,51]. Rural family medicine education lacks medical resources comparing to urban situations, so family medical teachers can teach critical skills as physicians through writing case reports for medical learners cognizance. Furthermore, in project-based learning, the skills and attitude toward collaboration can be nurtured through various collaborations and discussions with team members [52]. Learners can realize their responsibility as members to accomplish their projects [53]. As this study shows, participants realized their responsibility as researchers. This realization might lead to more importance being placed on scheduling and prompt communication with the co-authors so as not to delay the progression of writing case reports. This learning is essential for students’ prospective careers as medical doctors and scientists, and for preparing for the future [54,55]. In addition, rural family medicine education does not include fewer stakeholders and team members, and such situations are beneficial for establishing their relationship, compared to urban contexts. The established relationship between rural medical teachers and learners can facilitate case-report driven medical education effectively. Thus, case report-driven medical education can fit rural CBME regarding family medicine. Overall, writing case reports in PBME can be an effective way to teach and learn in the field of family medicine in the VUCA world. It can help students develop the skills and knowledge of medical scientists for their future careers, and enable them to apply their skills and knowledge in a real-world setting.” (Lines 376–431).

-          Student motivation affects learning on all subject matters. Do the authors think that student motivation might have introduced bias in this study? Or is the bias introduced by some students' willingness to participate in the study? The meaning of lines 390-391 is unclear.

-          Transferability and reliability are two different concepts.

-          What do the authors mean by ‘theoretical triangulation’?

Response:

We revised the limitation part comprehensively based on your suggestion, as follows:

“This study has several limitations. The first concerns the motivation of the participants regarding learning about family medicine and case reports, because the motivation of learning might affect the learning processes. The participants were motivated to learn about family medicine in community hospitals. The transferability to all medical students and residents may be problematic because not all are interested in writing case reports. In addition, this study was performed in only one rural community hospital. Thus, to overcome this limitation, the researchers clarified the learning content of multiple participants through iterative data collection and comprehensive descriptions of the contexts and learning methods. Another limitation is reliability; to improve reliability, we used iterative data analysis and a long duration of data collection. Future studies should investigate effective educational methods in other regions and in international contexts, including this study’s theory. Additionally, the first author coded the data transcripts, which could have affected the credibility of this study. To combat this, the second researcher reviewed the process of coding, concepts and themes for investigator triangulation.” (Lines 432–445).

Reviewer 3 Report

The research question “What and how do medical students and residents learn through case report-driven PBME in rural contexts?” (line 70-71) is extremely broad. To help determine the value of this question the following needs to occur in the introduction: 

A more robust discussion of the "learning" the authors are seeking to investigate. The word "learning" can encompass many, many things. What "learning" are the authors seeking to understand?

The research design is also extremely broad. The four questions were identified as:

1. What did you think of writing case reports?

2. How did you evaluate the effectiveness of writing case reports during your learning?

3. How do you evaluate the difficulty of writing case reports in your learning process?

4. Do you have any ideas on how to improve the quality of learning by writing case reports?

The relationship between some of these questions and the overall research question is problematic. For example, "What did you think of writing case reports?" or "How do you evaluate the difficulty of writing case reports in your learning process?" are questions that do not directly address the research question about learning from writing case reports. Rather they are student self evaluations of learning - perhaps a useful measure but this needs to be discussed in the methods. 

Having said the above, the grounded theory approach to this study is relevant to a broad concept of "learning" and it may be useful to revise the paper to reflect this broader approach. I would suggest two  elements of literature that should be included: 

1. More discussion of what is meant by "learning" in medical education. 

2. The value of student self-evaluation. Because the study questions rely on student's perceptions of learning from report writing, it would be useful to justify this as a method of data collection. 

I would also suggest deleting Table 1 as I did not find it useful. While I found  section 2.3 extremely helpful, I was still confused about the actual structure of the case reporting and would add the template of the case report structure (including any instructions in that template).

Overall, I found the article clear, well-written, and easy to follow. 

Author Response

The research question “What and how do medical students and residents learn through case report-driven PBME in rural contexts?” (line 70-71) is extremely broad. To help determine the value of this question the following needs to occur in the introduction: 

Response:

We concur, and revised the research question and last paragraph as follows:

“Case report-driven medical education is particularly useful in resource-limited medical education, such as rural community-based medical education (CBME), which suffers from a lack of medical teachers and teaching resources [28–30]. In addition, learning from writing case reports of complicated older patients in rural contexts could enhance understanding of varieties of older patients’ presentations such as VUCA (volatility, uncertainty, complexity, and ambiguity), describing challenging situations in caring for older people. However, case report-driven medical education has not been investigated adequately in contemporary medical education. Therefore, we posed the research question: “What do medical students and residents learn about patients’ care and physician’s academic careers through case report-driven PBME in rural community hospitals?” Clarification of specific case report-driven medical education methods and learning outcomes in resource-limited settings can motivate medical educators in rural contexts to adopt PBME and case reports, thereby benefiting rural CBME. Therefore, this study aimed to clarify the specific methods and learning outcomes of case report-driven medical education in rural CBME.” (Lines 69–83).

A more robust discussion of the "learning" the authors are seeking to investigate. The word "learning" can encompass many, many things. What "learning" are the authors seeking to understand?

Response:

We revised the discussion part comprehensively, including benefits for case report-driven medical education in rural contexts:

“Case report-driven medical education, based on the results of the perception of medicine, can prepare medical students and residents for the VUCA world that is emerging in medicine. VUCA refers to volatile, uncertain, complex, and ambiguous conditions, expressing the present medical conditions across the world [38,39]. After the COVID-19 pandemic, education in the VUCA world has been emphasized in medical education, and applied to geriatric care [40]. In this study, medical students and junior residents learned the concepts of VUCA by writing case reports of complicated older patients in a rural community hospital. Medical students and residents had to consider complicated backgrounds, volatile conditions, clinical course uncertainty, and decision-making ambiguity when writing case reports of the complicated cases that they had not encountered in the urban general and university hospitals. Writing the case reports required active involvement from medical students and residents, and provided opportunities to consider VUCA aspects of geriatric care in rural hospitals [41,42]. Family medicine education can also use the concept of VUCA in educating challenging situations of the care of elderly patients with multimorbidity [43]. This study is the first to show the effect of project-based medical education using case report writing to learn the concept of VUCA in rural contexts.

An academic career as a medical professional can be beneficial for medical students and residents seeking to enhance their professional careers. Academic research on the compilation of clinical experience with scientific data references begins with writing case reports, which requires systematic consideration of patient conditions [44,45]. In this study, writing case reports allowed participants to realize the importance of compiling their experiences and sharing evidence with other medical professionals. Through the discussion of case reports with family medicine residents and teachers, the participants realized the importance of systems thinking in geriatric patient care and patient-centered methods, leading to an interest in family medicine. Furthermore, learners’ implementation of evidence-based medicine (EBM) can become realistic through the experience of writing case reports of complicated older patients. Previous studies have shown that EBM in geriatric care includes VUCA factors and requires shared decision-making (SDM) with patients and their siblings [46,47]. SDM learning can be promoted through concrete cases and discussions with peers [48,49]. As this study shows, writing case reports can provide participants with opportunities to apply EBM with SDM to the uncertainty of geriatric care.

Writing case reports can facilitate various ways of cooperation among medical students, residents, and teachers, enhancing trainees’ responsibility in collaboration. Improved relationships among them could be enhanced through writing case reports. Writing case reports is a type of PBME shown to be an effective way to teach and learn critical thinking and problem-solving, as exemplified by this study’s other themes [50,51]. Rural family medicine education lacks medical resources comparing to urban situations, so family medical teachers can teach critical skills as physicians through writing case reports for medical learners cognizance. Furthermore, in project-based learning, the skills and attitude toward collaboration can be nurtured through various collaborations and discussions with team members [52]. Learners can realize their responsibility as members to accomplish their projects [53]. As this study shows, participants realized their responsibility as researchers. This realization might lead to more importance being placed on scheduling and prompt communication with the co-authors so as not to delay the progression of writing case reports. This learning is essential for students’ prospective careers as medical doctors and scientists, and for preparing for the future [54,55]. In addition, rural family medicine education does not include fewer stakeholders and team members, and such situations are beneficial for establishing their relationship, compared to urban contexts. The established relationship between rural medical teachers and learners can facilitate case-report driven medical education effectively. Thus, case report-driven medical education can fit rural CBME regarding family medicine. Overall, writing case reports in PBME can be an effective way to teach and learn in the field of family medicine in the VUCA world. It can help students develop the skills and knowledge of medical scientists for their future careers, and enable them to apply their skills and knowledge in a real-world setting.” (Lines 376–430).

The research design is also extremely broad. The four questions were identified as:

  1. What did you think of writing case reports?
  2. How did you evaluate the effectiveness of writing case reports during your learning?
  3. How do you evaluate the difficulty of writing case reports in your learning process?
  4. Do you have any ideas on how to improve the quality of learning by writing case reports?

The relationship between some of these questions and the overall research question is problematic. For example, "What did you think of writing case reports?" or "How do you evaluate the difficulty of writing case reports in your learning process?" are questions that do not directly address the research question about learning from writing case reports. Rather they are student self evaluations of learning - perhaps a useful measure but this needs to be discussed in the methods. 

Response:

We agree, and revised the translation of the four questions from Japanese to English as follows, based on the revision of the research question:

“The interview guide included four questions: What did you think of the experience of writing case reports? What did you learn through writing case reports in respect of patient management? What did you learn through writing case reports regarding physicians’ careers? Do you have any ideas on learning by writing case reports? Each interview lasted approximately 30 minutes.The semi-structured interviews and the field notes regarding case report-driven medical education were used in the analysis.” (Lines 156–162).

Having said the above, the grounded theory approach to this study is relevant to a broad concept of "learning" and it may be useful to revise the paper to reflect this broader approach. I would suggest two  elements of literature that should be included: 

  1. More discussion of what is meant by "learning" in medical education. 
  2. The value of student self-evaluation. Because the study questions rely on student's perceptions of learning from report writing, it would be useful to justify this as a method of data collection. 

Response:

We revised the description of grounded theory approaches as reflected below. Regarding the VUCA concepts, we described the process and the concept in the analysis section (Lines 179–181).

“The grounded theory approach was used to clarify the participants’ learning through case report-driven medical education [36]. All the field notes and interviews were recorded and transcribed verbatim. After reviewing the field notes and conducting the first five semi-structured interviews, the first researcher, R.O., coded the data and developed codebooks based on the repeated reading of field notes as initial coding for reliability. This study used process and concept coding methods [37]. The coded contents in the initial coding were shared with the second researcher, C.S., and the two researchers discussed the contents for triangulation. Then, R.O. continued initial coding individually, creating tentative concepts. The interview contents were analyzed iteratively during the research period after finishing each participant’s case report learning for theoretical saturation. After the initial coding and creating tentative concepts, R.O. induced, merged, deleted, and refined the tentative concepts and created themes by going back and forth between the research materials and initial coding for axial coding. During the code refinement process, axial coding was used to elaborate on tentative concepts and themes. R.O. and C.S. discussed the concepts and themes created through axial coding for further triangulation. Through the discussion, the concept of VUCA (volatility, uncertainty, complexity, ambiguity) appeared, which was incorporated into the themes and concepts regarding the previous article on the present medical education [38]. Finally, the theory, comprising concepts and themes, was discussed by both researchers, who ultimately agreed on the final themes.” (Lines 164–183).

I would also suggest deleting Table 1 as I did not find it useful. While I found  section 2.3 extremely helpful, I was still confused about the actual structure of the case reporting and would add the template of the case report structure (including any instructions in that template).

Response:

Thank you for your productive feedback. We have agreed with the feedback and revised the section 2.2 and 2.3 comprehensively including the participants’ background and precise information of case report writing as follows.

“A total of 45 medical students (six 4th-grade, 16 5th-grade and 23 6th-grade participants), seven junior residents, and eight family medicine residents participated in this research. The female percentage was 43.3% (26/60). Finally, fifty-one case reports were completed and published in the international medical journal, Cureus, from April 2021 to March 2023 (Table 1). The described patients’ average age was 74.5 years old (standard deviation = 18.1), and 47.1 % were male.” (Lines 192–197).

“Case report-driven medical education began in the first week of rural CBME training, during which a family medicine teacher explained case report-driven medical education. 

Medical students examined cases with family medicine residents and educators, taking clinical histories and noting physical examinations. Based on these clinical notes, they discussed the patients’ clinical conditions. In some cases, medical educators provided several articles and learning materials to medical students and junior residents to facilitate their understanding of patients’ conditions. Based on the discussions and relevant reading material, medical students and residents progressed in writing their case reports. Cases of patients actively infected with COVID-19 were not allocated to participants of this study because of the risk of infections and the limitation of interaction with patients.

The case reports comprised four sections: introduction, case presentation, discussion, and conclusions. The writing was facilitated by discussions among medical students, junior residents, family medicine residents, and teachers. Case selection was performed through discussions with the participants. First, medical students and junior residents summarized patients’ diseases and specific features. Residents and teachers of family medicine reviewed and revised these descriptions. Second, the medical students and junior residents described the patients’ clinical histories, physical examinations, specific tests, and treatments. When the medical students and junior residents struggled with their descriptions, they discussed them with family medicine residents and teachers. They re-examined the patients with family medicine residents or medical teachers to supplement their documentation.

Third, the medical students and residents described the discussion and conclusions based on what they had learned. Through discussions with family medicine residents and teachers, they selected topics and started to draft their discussion sections. Family medicine residents and teachers reviewed the medical students’ descriptions, and revised the content regarding paragraphs and academic writing. All documentation for the case reports was provided by medical students, junior residents, family medicine residents, and medical teachers. Finally, the family medicine residents and medical teachers reviewed and revised the content of the case reports and submitted each manuscript to an international medical journal, Cureus.” (Lines 117–146).

Overall, I found the article clear, well-written, and easy to follow. 

Round 2

Reviewer 3 Report

The authors have effectively addressed my identified concerns with the first manuscript in this revision. I feel these changes add to the relevance and interest of the paper and therefore recommend publication with no further required changes.